# Solubility of Sulfamethazine in Acetonitrile–Ethanol Cosolvent Mixtures: Thermodynamic Analysis and Mathematical Modeling

**DOI:** 10.3390/molecules30173590

**Published:** 2025-09-02

**Authors:** Diego Ivan Caviedes-Rubio, Cristian Buendía-Atencio, Rossember Edén Cardenas-Torres, Claudia Patricia Ortiz, Fleming Martinez, Daniel Ricardo Delgado

**Affiliations:** 1Programa de Ingeniería Agroalimentaria, Grupo de Investigación de Ingenierías UCC-Neiva, Facultad de Ingeniería, Universidad Cooperativa de Colombia, Sede Neiva, Calle 11 No. 1-51, Neiva 410001, Colombia; diego.caviedesr@campusucc.edu.co; 2Facultad de Ciencias, Departamento de Química, Universidad Antonio Nariño, Bogotá 110231, Colombia; c.buendia@uan.edu.co; 3Grupo de Energía Materiales y Diseño EnerDIMAT, Facultad de Ingeniería, Universidad de América, Av. Circunvalar No. 20-53, Bogotá 110321, Colombia; rossember.cardenas@uamerica.edu.co; 4Ingeniería Industrial, Institución Universitaria Politécnico Grancolombiano, Bogotá 110321, Colombia; cportizd14@gmail.com; 5Grupo de Investigaciones Farmacéutico-Fisicoquímicas, Departamento de Farmacia, Facultad de Ciencias, Universidad Nacional de Colombia, Sede Bogotá, Carrera 30 No. 45-03, Bogotá 110321, Colombia; fmartinezr@unal.edu.co; 6Programa de Ingeniería Civil, Grupo de Investigación de Ingenierías UCC-Neiva, Facultad de Ingeniería, Universidad Cooperativa de Colombia, Sede Neiva, Calle 11 No. 1-51, Neiva 410001, Colombia

**Keywords:** sulfamethazine, cosolvent, solubility, Gibbs–van’t Hoff–Krug model

## Abstract

The low water solubility of sulfamethazine (SMT) limits its clinical efficacy, making it crucial to study techniques such as cosolvency to optimize pharmaceutical formulations. This study aimed to thermodynamically evaluate the solubility of SMT in {acetonitrile (MeCN) + ethanol (EtOH)} cosolvent mixtures over a temperature range of 278.15 to 318.15 K in order to understand the molecular interactions that govern this process. SMT solubility in the mixtures was measured using a flask-shaking method. The solid phases were analyzed using differential scanning calorimetry (DSC) to rule out polymorphisms. Using the Gibbs–van’t Hoff–Krug model, we calculated the apparent thermodynamic functions of the solution and mixture from the obtained data. The results showed that solubility increased almost linearly with MeCN fraction and temperature, indicating that MeCN is a more efficient solvent and that the process is endothermic. Thermodynamic analysis revealed that dissolution is an endothermic process with favorable entropy for all compositions. The higher solubility in MeCN is attributed to the lower energetic cost required to form the solute cavity compared to the high energy needed to disrupt the hydrogen bond network of ethanol. This behavior can be explained by an enthalpy–entropy compensation phenomenon. This phenomenon provides an essential physicochemical basis for designing pharmaceutical processes.

## 1. Introduction

Sulfamethazine (SMT) (Figure 1) is a sulfonamide antibiotic widely used in veterinary medicine to prevent and treat bacterial and protozoal infections [1,2,3,4]. However, the low water solubility of SMT poses a significant challenge in developing pharmaceutical formulations, which directly affects its bioavailability and clinical efficacy [5,6,7,8]. Therefore, solubility studies are a fundamental pillar of pharmaceutical science because they provide essential physicochemical data for designing drug purification, crystallization, and dosing processes [9,10]. One of the most effective and widely used techniques in industry to overcome this limitation is cosolvency, which increases the solubility of a solute by adding another solvent to a main solvent [11]. Ethanol and acetonitrile are important organic solvents in the pharmaceutical industry [12,13,14]. Ethanol is one of the most widely used solvents in the pharmaceutical industry. Its intermediate polarity enables it to dissolve a variety of active ingredients, particularly those that are insoluble in water. This facilitates the formulation of elixirs, tinctures, and injectable solutions, ensuring accurate dosing and optimal bioavailability. Additionally, ethanol is used to extract bioactive compounds from natural sources [15]. In this context, ethanol is an indispensable tool that ensures the efficacy, safety, and stability of countless medications [16]. Acetonitrile, on the other hand, is one of the most widely used organic solvents in quality control and industrial synthesis. It is used primarily as the preferred mobile phase in high-performance liquid chromatography (HPLC) due to its low absorbance in the UV spectrum, which allows for the sensitive detection of analytes, its low viscosity, which ensures efficient separation with less back pressure in the system, and its miscibility with water, which creates the necessary elution gradients [17]. Acetonitrile is also used in the synthesis of active ingredients because it can dissolve polar intermediates and reagents without interfering with nucleophilic reactions [18]. They are used extensively in synthesis, recrystallization, and chromatographic purification methods due to their favorable physicochemical properties and ability to modulate the polarity of the medium [19].

Despite their industrial importance, these solvents have a gap in the scientific literature regarding systematic data on sulfametazine solubility in cosolvent mixtures of MeCN and ethanol at different temperatures. Understanding how SMT behaves in these ternary systems (SMT + MeCN + EtOH) is crucial for optimizing industrial processes, such as crystallization, where the precise control of solubility is key to maximizing the yield and purity of the active ingredient. Beyond empirical data, evaluating the apparent thermodynamic parameters of the solution process (Gibbs energy, enthalpy, and entropy) and developing predictive mathematical models are essential to minimizing experimentation and efficiently scaling up processes [21]. The main objective of this research is therefore to thermodynamically evaluate the solubility of sulfametazine in MeCN–ethanol cosolvent mixtures from 278.15 K to 318.15 K.

This research presents the mole fraction solubility data of SMT, calculates and analyzes the apparent thermodynamic functions of dissolution, and correlates the experimental data using mathematical models. This study is necessary to generate fundamental data and develop robust mathematical models that can be directly applied in the pharmaceutical industry for the design and optimization of sulfametazine-related processes. These results will provide a comprehensive understanding of the thermodynamics of this system and offer the industry a valuable tool for research and development processes.

## 2. Results and Discussion

Table 1 reports the experimental solubility data of SMT in pure MeCN and EtOH, as well as in 19 {MeCN (1) + EtOH (2)} cosolvent mixtures, at nine temperatures. These data provide a graphical representation of the dissolution process. Figure 2 shows the solubility of SMT as a mole fraction ($x_{3}$) on a logarithmic scale as a function of the mass fraction of MeCN ($w_{1}$). The graph shows isotherms ranging from 278.15 to 318.15 K and illustrates two key thermodynamic phenomena that govern the dissolution process.

A positive effect of the solvent composition can be observed. At constant temperature, the solubility of SMT increases nearly linearly as the mass fraction of MeCN increases from 0 (pure EtOH) to 1 (pure MeCN). This behavior indicates that MeCN is a significantly more effective solvent for SMT than EtOH. The molecular explanation is based on the physicochemical properties of the solute and solvents. SMT is an amphiphilic molecule with a $\log k_{OW}$ of approximately 1 (0.89) [22,23,24]. This indicates a balance between its lipophilic character, represented by the benzene ring, and its polar character, represented by the sulfonamide group and pyrimidine heterocycles. SMT is also amphiprotic, which is a fundamental characteristic for analyzing its behavior in solution. Regarding its hydrogen bonding capacity, SMT has three proton donor sites: the two hydrogens of the aniline amino group (${-\mathrm{NH}}_{2}$) and the single hydrogen of the sulfonamide group −NH−. It also has multiple acceptor sites, including the two oxygen atoms of the sulfonyl group, the two nitrogen atoms of the pyrimidine ring, and the nitrogen atom of the aniline group. In this context, EtOH can act as a hydrogen bond acceptor for the acidic protons of SMT (SMT (${-\mathrm{NH}}_{2}$) … (O−H) EtOH), and simultaneously, it can act as a hydrogen bond donor to the multiple acceptor sites of SMT (such as the oxygens of the sulfonyl group and the nitrogens of the pyrimidine: SMT (−SO_2_−) … (O−H) EtOH). These interactions create the potential for a complex, multipoint hydrogen bond network between SMT and the surrounding EtOH molecules. As the concentration of MeCN increases, acid–base interactions may occur between the sulfonamide proton of SMT and the nitrogen atom of MeCN: SMT (${-\mathrm{SO}}_{2}$NH−) … (C−N) MeCN. This could be one of the most favorable solute–solvent interactions in the system and would explain the increase in SMT solubility when MeCN is added.

**Table 1 molecules-30-03590-t001:** Experimental solubility of sulfamethazine (SMT) ($10^{4}x_{3}$) in cosolvent mixtures {MeCN (1) + EtOH (2)} between 278.15 and 318.15 K (± are the standard deviations). Experimental pressure *p*: 0.096 MPa ^a^.

*w*_1_ ^c^	Temperature/K ^b^				
	278.15	283.15	288.15	293.15	298.15
0.00	3.84 ± 0.033	4.89 ± 0.05	6.02 ± 0.01	7.47 ± 0.04 ^d^	9.18 ± 0.07 ^d^
0.05	4.14 ± 0.05	5.3 ± 0.05	6.52 ± 0.09	8.12 ± 0.04	9.94 ± 0.07
0.10	4.47 ± 0.05	5.7 ± 0.08	6.97 ± 0.07	8.53 ± 0.03	10.58 ± 0.13
0.15	4.85 ± 0.06	6.12 ± 0.09	7.6 ± 0.09	9.3 ± 0.14	11.47 ± 0.22
0.20	5.28 ± 0.04	6.59 ± 0.06	8.26 ± 0.07	10.04 ± 0.13	12.37 ± 0.19
0.25	5.62 ± 0.09	7.14 ± 0.04	8.83 ± 0.1	10.75 ± 0.14	13.4 ± 0.19
0.30	6.08 ± 0.10	7.68 ± 0.05	9.41 ± 0.14	11.42 ± 0.05	14.56 ± 0.07
0.35	6.52 ± 0.08	8.2 ± 0.03	10.15 ± 0.23	12.42 ± 0.16	15.72 ± 0.17
0.40	7.27 ± 0.14	9.15 ± 0.10	11.44 ± 0.03	13.96 ± 0.05	17.16 ± 0.24
0.45	7.57 ± 0.02	9.66 ± 0.18	11.99 ± 0.11	14.61 ± 0.15	18.13 ± 0.22
0.50	8.27 ± 0.11	10.46 ± 0.08	12.73 ± 0.08	15.69 ± 0.29	19.44 ± 0.36
0.55	8.96 ± 0.08	11.14 ± 0.10	13.93 ± 0.16	16.99 ± 0.17	21.11 ± 0.36
0.60	9.65 ± 0.12	12.06 ± 0.04	14.96 ± 0.12	18.07 ± 0.14	22.96 ± 0.21
0.65	10.62 ± 0.23	12.92 ± 0.12	16.08 ± 0.29	19.36 ± 0.21	24.6 ± 0.5
0.70	11.33 ± 0.23	14.04 ± 0.13	17.34 ± 0.27	20.94 ± 0.13	26.28 ± 0.3
0.75	12.38 ± 0.22	15.40 ± 0.06	18.6 ± 0.14	22.48 ± 0.28	28.73 ± 0.26
0.80	13.29 ± 0.26	16.51 ± 0.17	20.01 ± 0.25	24.58 ± 0.15	30.81 ± 0.08
0.85	14.48 ± 0.21	17.57 ± 0.30	21.95 ± 0.23	26.13 ± 0.4	33.35 ± 0.4
0.90	15.6 ± 0.1	19.03 ± 0.29	23.6 ± 0.3	28.18 ± 0.4	35.68 ± 0.4
0.95	16.51 ± 0.09	20.44 ± 0.20	25.11 ± 0.19	29.97 ± 0.06	38.49 ± 0.31
1.00	17.8 ± 0.35 ^e^	22.04 ± 0.11 ^e^	27.14 ± 0.13 ^e^	33.1 ± 0.32 ^e^	41.52 ± 0.3 ^e^
*w*_1_ ^c^	Temperature/K ^b^				
	303.15	308.15	313.15	318.15	
0.00	10.87 ± 0.14 ^c^	12.99 ± 0.23 ^d^	16.23 ± 0.16 ^d^	18.96 ± 0.15	
0.05	11.78 ± 0.09	13.95 ± 0.24	17.32 ± 0.09	20.51 ± 0.19	
0.10	12.74 ± 0.19	15.09 ± 0.13	18.77 ± 0.15	21.73 ± 0.07	
0.15	13.77 ± 0.19	16.1 ± 0.14	20.36 ± 0.14	23.59 ± 0.15	
0.20	14.67 ± 0.04	17.61 ± 0.08	21.98 ± 0.14	25.62 ± 0.34	
0.25	15.99 ± 0.19	18.63 ± 0.20	23.26 ± 0.21	27.76 ± 0.13	
0.30	17.36 ± 0.12	20.29 ± 0.4	25.17 ± 0.25	29.32 ± 0.33	
0.35	18.70 ± 0.15	21.7 ± 0.17	26.7 ± 0.07	31.82 ± 0.4	
0.40	20.66 ± 0.25	24.49 ± 0.15	30.01 ± 0.2	35.08 ± 0.5	
0.45	21.95 ± 0.15	25.49 ± 0.4	31.18 ± 0.4	36.61 ± 0.24	
0.50	23.40 ± 0.29	27.51 ± 0.13	33.69 ± 0.4	39.82 ± 0.32	
0.55	25.4 ± 0.5	29.61 ± 0.17	35.69 ± 0.24	43.12 ± 0.23	
0.60	27.25 ± 0.24	31.65 ± 0.24	39.1 ± 0.49	45.76 ± 0.8	
0.65	29.46 ± 0.16	34.34 ± 0.21	42.27 ± 0.36	49.29 ± 0.7	
0.70	31.71 ± 0.54	36.81 ± 0.4	45.24 ± 0.23	52.83 ± 0.6	
0.75	33.84 ± 0.24	40.43 ± 0.28	48.66 ± 0.13	57.7 ± 0.6	
0.80	36.9 ± 0.4	43.03 ± 0.5	52.14 ± 0.7	61.1 ± 0.7	
0.85	39.6 ± 0.5	46.09 ± 0.2	55.04 ± 0.05	66.53 ± 0.8	
0.90	43.0 ± 0.5	49.87 ± 0.5	59.91 ± 0.6	71.36 ± 1	
0.95	45.77 ± 0.13	53.49 ± 0.4	64.27 ± 1.1	76.53 ± 0.9	
1.00	50.8 ± 0.9 ^e^	58.12 ± 0.4 ^e^	69.95 ± 1.7	83.07 ± 0.6 ^e^	

^a^ Standard uncertainty in pressure *u*(*p*) = 0.001 MPa, ^b^ Standard uncertainty in temperature is *u*(*T*) = 0.05 K, ^c^ Mass fraction of MeCN in MeCN + EtOH cosolvent mixtures free of SMT, ^d^ From of Ref. [25], ^e^ From of Ref. [26].

Figure 2 shows the isotherms arranged from bottom to top according to increasing temperature. This indicates that the SMT dissolution process is endothermic, meaning solubility at a fixed composition increases with temperature. At the molecular level, energy input is required to overcome solute–solute and solvent–solvent interactions in the SMT crystal lattice. Increasing temperature favors this process. Figure 2 illustrates a system in which solubility increases with temperature and with the transition from a protic solvent, such as EtOH [27], to an aprotic solvent, such as MeCN [28]. This highlights the importance of solvent selection when optimizing the dissolution of drugs with similar physicochemical characteristics to sulfametazine.

Delgado et al. conducted a systematic study on the solubility of sulfadiazine (SD) [29], sulfamerazine (SMR) [30], and sulfamethazine (SMT) [26] in cosolvent mixtures of acetonitrile (MeCN), methanol (MeOH), ethanol (EtOH), and 1−propanol (1−PrOH) to identify possible relationships between the structure of the studied sulfonamides and alcohols and the variation in solubility. The results reveal a consistent and counterintuitive pattern regarding the solubility parameter: solubility is consistently lower in pure alcohol and higher in pure acetonitrile. This trend correlates directly with the acidity of the solvent as a hydrogen bond donor (parameter α [31]) (Figure 3a) rather than with the solubility parameter of the drugs and solvents (Figure 3b). This phenomenon can be explained by the acid–base behavior of the components and the energetics of cavity formation. SMT is an acid (sulfonamide proton) that seeks a basic environment for favorable interaction. MeCN is a moderate base (β = 0.40 [32]) and a weak acid (α = 0.19), providing an easily accessible basic site (nitrogen) without strong competitive interactions. Conversely, EtOH is a stronger base (β = 0.75 [32]) but also a stronger acid (α = 0.86 [31]). The high acidity of EtOH allows it to participate in self-association through hydrogen bonds [33,34]. Therefore, for SMT to dissolve in EtOH, it must break the highly stable and energetically favorable EtOH–EtOH hydrogen bond network. The energy required to break these solvent–solvent bonds is substantial. In contrast, MeCN has much weaker self-interactions, so creating a cavity in MeCN and allowing SMT to interact requires much less energy. This phenomenon can be observed when analyzing the solubility of SMT in three cosolvent mixtures: MeCN + MeOH, MeCN + EtOH, and MeCN + 1-PrOH (Figure 3c). With the exception of the MeCN + MeOH mixture, maximum solubility is achieved in pure MeCN. Comparing the solubility of SMT as a function of alcohol reveals that as the number of carbons in the aliphatic chain of the alcohol increases, SMT becomes less soluble, preferring MeCN + MeOH mixtures and exhibiting lower solubility in MeCN + 1-PrOH. This finding supports the theory that solubility depends on the energy of cavity formation. Another important comparison is analyzing the solubility of the three sulfonamides in the MeCN + EtOH mixture (Figure 3d). Similar to Delgado and Martínez’s studies on hydroethanolic mixtures, the solubility order of the three sulfonamides is SMT > SMR > SD [25]. This behavior may be related to the energetics of the solid state [35]. When analyzing properties such as the temperature and enthalpy of fusion of the three sulfonamides, solute–solute molecular interactions are lower in SMT and higher in SD. Therefore, it can be conjectured that SMT has greater solubility due to lower intermolecular energetics.

### 2.1. Calorimetric Analysis of SMT

Table 2 shows the melting temperatures and enthalpies of various crystalline SMT samples. The experimental values for the commercial sample agree with the reference data reported in the literature, thus verifying the identity and purity of the initial material. The sharpness of the endothermic peaks in Figure 4’s thermograms supports this conclusion since impurities generally lead to broadened and depressed melting peaks.

Because different crystalline forms of the same drug can exhibit different solubility, stability, and bioavailability properties, controlling polymorphism is a critical aspect of pharmaceutical development [36]. Thus, it was necessary to analyze the SMT recrystallized from different solvent systems (pure EtOH, pure MeCN, and a 50/50 mixture) to assess whether the process induces a polymorphic transition.

The melting points and enthalpies of all the recrystallized samples are remarkably similar to each other and to the original sample. The small variations observed are within the experimental uncertainty range and do not indicate a change in crystal structure. A true polymorphic change would result in much greater differences in both the melting temperature and the melting enthalpy. Therefore, the analysis demonstrates that the recrystallization of SMT from EtOH, MeCN, or mixtures thereof does not induce polymorphism. The drug maintains its stable crystalline form. This favorable result indicates that using these solvents in purification and quantification processes does not compromise the consistency of the API’s solid state.

**Table 2 molecules-30-03590-t002:** Enthalpy of fusion and melting temperature of SMT.

Sample	Enthalpy ^a^, $\Delta_{\mathrm{fus}}H/\mathrm{kJ}\cdot {\mathrm{mol}}^{-1}$	Temperature ^a^, $T_{\mathrm{fus}}/K$	Ref.
Original sample	33.41 ± 0.3	468.9 ± 0.5	This work
	33.57 ± 0.3	468.5 ± 0.5	[26]
		468.6	[37]
	31.1	471.7	[38]
	39.2	469	[39]
Acetonitrile	33.57 ± 0.3	467.9 ± 0.5	This work
	34.27 ± 0.3	468.7 ± 0.5	[26]
Acetonitrile ($w_{1}=0.50$)	32.27 ± 0.3	467.9 ± 0.5	This work
Ethanol	33.27 ± 0.3	469.7 ± 0.5	This work

^a^ ± are the standard deviations.

### 2.2. Thermodynamic Analysis of the Solubility of SMT in Cosolvent Mixtures {MeCN (1) + EtOH (2)}

From the SMT solubility data in {MeCN (1) + EtOH (2)} cosolvent mixtures, the apparent thermodynamic solution functions were calculated using the Gibbs–van’t Hoff–Krug model according to [26,40,41]:(1)$$\Delta_{\mathrm{soln}}H^{\circ }=-R{\left(\frac{\partial \ln x_{3}}{\partial (T^{-1}-T_{\mathrm{hm}}^{-1})}\right)}_{p}=-R\cdot m$$(2)$$\Delta_{\mathrm{soln}}G^{\circ }=-RT_{\mathrm{hm}}\cdot a$$(3)$$\Delta_{\mathrm{soln}}S^{\circ }=\left(\Delta_{\mathrm{soln}}H^{\circ }-\Delta_{\mathrm{soln}}G^{\circ }\right)T_{\mathrm{hm}}^{-1}$$(4)$$\zeta_{H}=|\Delta_{\mathrm{soln}}H^{\circ }\left|(\right|\Delta_{\mathrm{soln}}S^{\circ }\left|+\right|\Delta_{\mathrm{soln}}H^{\circ }{\left|\right)}^{-1}$$(5)$$\zeta_{TS}=1-\zeta_{H}$$
where $\Delta_{\mathrm{soln}}G^{\circ }$, $\Delta_{\mathrm{soln}}H^{\circ }$, $\Delta_{\mathrm{soln}}S^{\circ }$, and $T_{\mathrm{hm}}\Delta_{\mathrm{soln}}S^{\circ }$, are the Gibbs energy (in kJ·${\mathrm{mol}}^{-1}$), enthalpy (in kJ·${\mathrm{mol}}^{-1}$) and entropy (in J·${\mathrm{mol}}^{-1}$· $K^{-1}$) of the solution. $\zeta_{H}$, and $\zeta_{TS}$ describe the contribution of the energetic and organizational components to the value of the Gibbs energy of solution. As for $T_{\mathrm{hm}}$, called harmonic mean temperature, it is the harmonic mean of the study temperatures [26].

From the van’t Hoff–Krug equation, *m* and *a* are calculated, from which $\Delta_{\mathrm{soln}}G^{\circ }$ and $\Delta_{\mathrm{soln}}H^{\circ }$ are calculated according to Equations (Equation 1) and (Equation 2) [26].(6)$$\ln x_{3}=m\cdot \left(T^{-1}-T_{\mathrm{hm}}^{-1}\right)+a$$

Table 3 shows the thermodynamic solution functions of SMT in {MeCN (1) + EtOH (2)} cosolvent mixtures. The Gibbs free energy of solution ($\Delta_{\mathrm{soln}}G^{\circ }$) is positive in all compositions and decreases from 17.39 kJ·${\mathrm{mol}}^{-1}$ in pure EtOH to 13.66 kJ·${\mathrm{mol}}^{-1}$ in pure MeCN. This decrease in Gibbs energy directly correlates with the observed increase in SMT’s solubility as the mixture becomes enriched in MeCN, indicating a greater affinity of the solute for solvents richer in MeCN. When breaking down the driving forces, the enthalpic and entropic contributions are examined. The solution enthalpy ($\Delta_{\mathrm{soln}}H^{\circ }$) is positive across the entire range of compositions, confirming that the process is endothermic. This indicates that a net energy input is necessary to overcome cohesive interactions, primarily the energy of the solute crystal lattice (SMT–SMT interactions) and the energy of solvent–solvent interactions (primarily strong hydrogen bonds in the EtOH lattice). At the molecular level, the slight decrease in enthalpy (from 29.21 to 28.5 kJ·${\mathrm{mol}}^{-1}$) as the MeCN fraction increases indicates a more favorable energy balance in the latter. However, it is important to note that the energy difference falls within the range of experimental error. Therefore, this result is not conclusive. With regard to entropic change, ($\Delta_{\mathrm{soln}}S^{\circ }$) is positive and increases with the MeCN fraction, ranging from 39.7 to 50 J·${\mathrm{mol}}^{-1}$$\cdot K^{-1}$. A positive entropy value is thermodynamically favorable, which may occur possibly because the solute and solvent molecules have a greater number of possible configurations compared to their initial state (solid solute and pure solvents). In addition, the addition of MeCN to the system may increase the destructuring of ethanol aggregates.

Regarding the contribution of enthalpy ($\zeta_{H}$) and entropy ($\zeta_{TS}$), $\zeta_{H}$ is greater than 0.66 in all cases, indicating a greater contribution of the energy component to the Gibbs energy of the solution [42]. Figure 5 shows a Perlovich diagram for the dissolution process of sulfametazine (SMT) in cosolvent mixtures of MeCN and EtOH [43]. The diagram allows one to analyze the thermodynamic functions that govern the solution process by representing the entropic contribution to Gibbs energy ($T\Delta_{\mathrm{soln}}S^{\circ }$) versus solution enthalpy ($\Delta_{\mathrm{soln}}H^{\circ }$). Thus, values located in sectors I, IV, V, and VIII indicate enthalpic conduction and values located in sectors II, III, VI, and VII indicate entropic conduction, which is in accordance with the following:Sector I: $\Delta_{\mathrm{soln}}H^{\circ }>T\Delta_{\mathrm{soln}}S^{\circ }$;Sector II: $T\Delta_{\mathrm{soln}}S^{\circ }>\Delta_{\mathrm{soln}}H^{\circ }$;Sector III: $\Delta_{\mathrm{soln}}H^{\circ }<0;T\Delta_{\mathrm{soln}}S^{\circ }>0;|T\Delta_{\mathrm{soln}}S^{\circ }\left|>\right|\Delta_{\mathrm{soln}}H^{\circ }|$;Sector IV: $\Delta_{\mathrm{soln}}H^{\circ }<0;T\Delta_{\mathrm{soln}}S^{\circ }>0;|\Delta_{\mathrm{soln}}H^{\circ }|>|T\Delta_{\mathrm{soln}}S^{\circ }|$;Sector V: $\Delta_{\mathrm{soln}}H^{\circ }<0;T\Delta_{\mathrm{soln}}S^{\circ }<0;|\Delta_{\mathrm{soln}}H^{\circ }|>|T\Delta_{\mathrm{soln}}S^{\circ }|$;Sector VI: $\Delta_{\mathrm{soln}}H^{\circ }<0;T\Delta_{\mathrm{soln}}S^{\circ }<0;|T\Delta_{\mathrm{soln}}S^{\circ }\left|>\right|\Delta_{\mathrm{soln}}H^{\circ }|$;Sector VII: $\Delta_{\mathrm{soln}}H^{\circ }>0;T\Delta_{\mathrm{soln}}S^{\circ }<0;|T\Delta_{\mathrm{soln}}S^{\circ }\left|>\right|\Delta_{\mathrm{soln}}H^{\circ }|$;Sector VIII: $\Delta_{\mathrm{soln}}H^{\circ }>0;T\Delta_{\mathrm{soln}}S^{\circ }<0;|\Delta_{\mathrm{soln}}H^{\circ }|>|T\Delta_{\mathrm{soln}}S^{\circ }|$.

**Figure 5 molecules-30-03590-f005:**
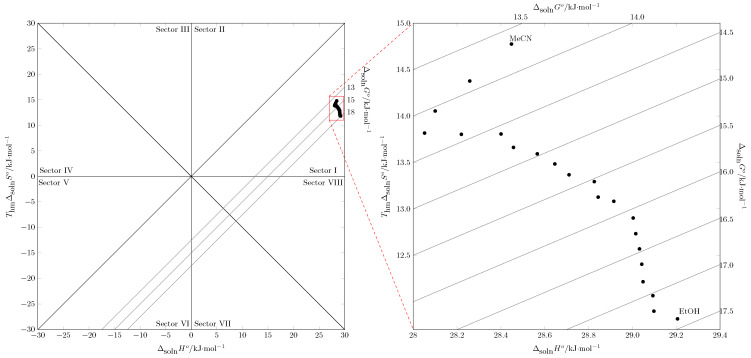
Perlovich´s plot for the solution process of SMT (3) in cosolvent mixtures {MeCN (1) + 1 − EtOH (2)} at $T_{\mathrm{hm}}$/K = 297.6. The gray lines represent the Gibbs energy of solution.

In this case, all points are located in sector I.($\Delta_{\mathrm{soln}}H^{\circ }>T\Delta_{\mathrm{soln}}S^{\circ }$), which corresponds to processes where the magnitude of the enthalpy is greater than the entropic contribution [44,45]. Since enthalpy and entropy are both positive for the dissolution process, this confirms that the phenomenon is fundamentally entropy-driven with a predominance of energetic contribution. At the molecular level, this implies that the main energy barrier to overcome is breaking cohesive interactions, such as solute–solute interactions in the SMT crystal lattice and solvent–solvent interactions, like the EtOH hydrogen bond network. Although the solution process generates a favorable increase in entropy, this effect is insufficient to counteract the enthalpy cost. This explains why the Gibbs energy of the solution is positive in all cases.

### 2.3. Thermodynamic Functions of Mixing of SMT in Cosolvent Mixtures {MeCN (1) + EtOH (2)}

The dissolution process can be understood as a sequence of several stages. The first consists of the hypothetical melting of the solute, transforming it into a supercooled liquid at the studied temperature, which is lower than its actual melting point. Subsequently, the solvent molecules must reorganize to create a cavity of the appropriate size to accommodate the solute. Finally, the solute molecule (in its supercooled liquid state) is introduced into this cavity. This last step is known as the mixing process, and its energy contribution to the overall process can be calculated as shown below [46,47]:(7)$$\Delta_{\mathrm{soln}}H^{o}=\Delta_{\mathrm{mix}}H^{o}+\Delta_{\mathrm{fus}}H^{T_{\mathrm{hm}}}$$(8)$$\Delta_{\mathrm{soln}}S^{o}=\Delta_{\mathrm{mix}}H^{o}+\Delta_{\mathrm{fus}}S^{T_{\mathrm{hm}}}$$
where $\Delta_{\mathrm{fus}}H^{T_{\mathrm{hm}}}$ and $\Delta_{\mathrm{fus}}S^{T_{\mathrm{hm}}}$ are the thermodynamic functions corresponding to the hypothetical melting process of the drug at $T_{\mathrm{hm}}$ = 297.6 K; these quantities are replaced by the enthalpy and entropy values of the ideal dissolution process, $\Delta_{\mathrm{soln}}H^{o-\mathrm{id}}$ and $\Delta_{\mathrm{soln}}S^{o-\mathrm{id}}$, which are taken from the literature [26].

The thermodynamic mixing functions are reported in Table 4. The Gibbs energy of mixing ($\Delta_{\mathrm{mix}}G^{\circ }$) is positive across the entire range of compositions, and it decreases from 6.1 kJ·${\mathrm{mol}}^{-1}$ in pure EtOH to 2.37 kJ·${\mathrm{mol}}^{-1}$ in pure MeCN. This trend demonstrates that solute–solvent interactions become progressively more favorable as the concentration of MeCN increases with this being the main factor driving the increased solubility of the drug in MeCN-rich mixtures and pure MeCN. The positive mixing enthalpy ($\Delta_{\mathrm{mix}}H^{\circ }$) reveals that the process is endothermic. At the molecular level, this result is primarily due to the energy required to form the cavity that accommodates the SMT molecule within the solvent structure. In EtOH, this cost is particularly high due to the need to break its hydrogen bond network. Since the intermolecular interactions of MeCN (induced dipole–induced dipole) are weaker than those of EtOH (dipole–dipole via hydrogen bonding), increasing the concentration of MeCN in the cosolvent mixture reduces the energy required to form the cavity. Therefore, the $\Delta_{\mathrm{mix}}H^{\circ }$ values tend to decrease. On the other hand, once the cavity is formed, solute–solvent interactions contribute to the decrease in enthalpy of mixing. Considering the dipole moment (12 D) and high polarizability (28.8 ${\overset{˚}{A}}^{3}$ [48,49]) of sulfametazine (SMT), a significant dipole would be induced in the surrounding solvent molecules (3.9 D and 3.4 ${\overset{˚}{A}}^{3}$ for MeCN and 1.7 D and 1.4 ${\overset{˚}{A}}^{3}$ for EtOH [50]), creating a strong attraction. SMT’s high polarizability also means that polar solvents’ electric fields will induce a dipole in it. Thus, the energy released by solute–solvent interactions dominated by polarization (induced dipole–dipole), especially with MeCN (in MeCN-rich mixtures and pure MeCN), would also contribute to the decrease in mixing enthalpy. Regarding the change in mixture entropy ($T\Delta_{\mathrm{mix}}S^{\circ }$), it is negative in EtOH-rich cosolvent mixtures (e.g., −5.9 J·${\mathrm{mol}}^{-1}\cdot K^{-1}$) in pure EtOH) and positive in MeCN-rich mixtures (e.g., 4 J·${\mathrm{mol}}^{-1}\cdot K^{-1}$) in pure MeCN), with a transition around a MeCN mass fraction of 0.65. From an entropic point of view, there is an entropic favoring of the mixing process from $w_{1}$ = 0.65 to pure MeCN. This change may be driven by acetonitrile breaking down the structure of ethanol, thus facilitating sulfametazine’s entry into the solution.

Figure 6 shows Perlovich’s diagram of the SMT mixing process in acetonitrile (MeCN) and ethanol (EtOH) mixtures. The trajectory of the experimental points through the different sectors of the diagram is revealing. In pure EtOH and EtOH-rich mixtures (in the lower right corner of the data set), the points are located in sectors I ($\Delta_{\mathrm{mix}}H^{\circ }>T\Delta_{\mathrm{mix}}S^{\circ }$) and (VIII $\Delta_{\mathrm{mix}}H^{o}>0;T\Delta_{\mathrm{mix}}S^{o}<0;|\Delta_{\mathrm{mix}}H^{o}|>|T\Delta_{\mathrm{mix}}S^{o}|$). In these sectors, the mixing process is dominated by enthalpy [43,44,45]. At the molecular level, the main energy barrier is positive enthalpy. This barrier is unfavorable to the mixing process and is associated with the high cost of forming a cavity in a strongly cohesive network of EtOH hydrogen bonds. The enthalpic cost exceeds the entropic contribution, which is unfavorable (negative) in these mixtures due to the structuring and ordering of the EtOH molecules around the solute.

### 2.4. Enthalpic Compensation of the SMT Solution Process in {MeCN (1) + EtOH (2)} Cosolvent Mixtures

The dissolution process is governed by fundamental equation $\Delta_{\mathrm{soln}}G^{\circ }=\Delta_{\mathrm{soln}}H^{\circ }-T\Delta_{\mathrm{soln}}S^{\circ }$ [51,52,53]. Figure 7 shows that as dissolution becomes less favorable ($\Delta_{\mathrm{soln}}G^{o}$ increases from 13.8 to 17.4 kJ·mol), the enthalpy of solution does not follow a linear trend [54,55]. Rather, it first decreases almost linearly from pure EtOH to $w_{1}$ = 0.85, reaching a minimum, and then increases to pure MeCN. This indicates that changes in enthalpy ($\Delta_{\mathrm{soln}}H^{\circ }$) and entropy ($T\Delta_{\mathrm{soln}}S^{\circ }$) are coupled rather than independent [56]. An unfavorable change from an enthalpic point of view is partially “compensated” by a favorable change from an entropic point of view and vice versa. According to Bustamante, the solution process is driven by enthalpy from pure EtOH up to $w_{1}$= 0.85 and by entropy of the solution from this composition up to pure MeCN [57,58]. From a molecular perspective, adding MeCN to EtOH makes the process progressively less endothermic ($\Delta_{\mathrm{soln}}H^{\circ }$ decreases). This is mainly because dissolving sulfamethazine requires breaking the organized structure of EtOH, which is held together by a network of hydrogen bonds. Creating a cavity in the EtOH to accommodate the solute requires more energy (it is a more endothermic process), so adding MeCN to the system favors the generation of the cavity, favoring the sulfamethazine solution process in terms of Gibbs energy (lower $\Delta_{\mathrm{soln}}G^{\circ }$).

## 3. Computational Validation

The van’t Hoff–Yalkowsky–Roseman model is one of the predictive solubility models widely used in pharmaceutical sciences to estimate the solubility of a solute in different cosolvent mixtures and at different temperatures. This semi-empirical approach combines two theories to provide more robust and comprehensive predictions based on four experimental solubility data points (two temperatures in two pure solvents).

The van’t Hoff equation focuses on the effect of temperature on solubility. The equation relates the logarithm of solubility in mole fraction ($\ln x_{3}$) to the inverse of the absolute temperature (1/*T*) [59,60].(9)$$\ln x_{3,i}=A\cdot \frac{1}{T}+B$$
where $x_{3}$ is the molar fraction solubility of the solute, *A* is the enthalpy of the solution coefficient, *B* is the entropy of the solution coefficient, and *T* is the absolute temperature in Kelvin.

The other component of the model is the Yalkowsky–Roseman equation [61,62,63]:(10)$$\ln x_{3,1+2}=w_{1}\ln x_{3,1}+w_{2}\ln x_{3,2}$$
where $x_{3}$ is the solubility of the drug in a cosolvent mixture, $w_{1}$ is the mass fraction of solvent 1 (less polar), $w_{2}$ is the mass fraction of solvent 2 (more polar), $x_{3,1}$ is the solubility of the drug in pure solvent 1, and $x_{3,2}$ is the solubility of the drug in pure solvent 2.

Combining the two models gives(11)$$\ln x_{3,1+2}=w_{1}\left(\frac{A_{1}}{T}+B_{1}\right)+w_{2}\left(\frac{A_{2}}{T}+B_{2}\right)$$

The mean relative deviation (MRD) was used to evaluate the variability of the model’s data in relation to the experimental data.(12)$$MRD=100\sum \left[\left(x_{3}^{E}-x_{3}^{C}\right)/x_{3}^{E})\right]N^{-1}$$
where $x_{3}^{E}$ is the experimental data, $x_{3}^{C}$ is the calculated data and *N* is the number of data in each set.

The resulted equation for the solubility prediction of SMT at different temperatures and composition ({MeCN (1) + EtOH (2)} ) is as follows:(13)$$\ln x_{3,1+2}=w_{1}\left(\frac{-3408.7}{T}+5.9236\right)+w_{2}\left(\frac{-3531.1}{T}+4.8307\right)$$

When evaluating the model using the mean relative deviation (MRD) across 189 data points, the overall MRD was found to be 1.31%. This prediction error falls within an acceptable range, as a model with an MRD value below 30% is considered promising in pharmaceutical sciences [64,65]. The primary advantage of the proposed model is that it does not require experimental solubility data from mixed solvents.

Figure 8 illustrates the correlation between the experimental and calculated data. Linear regression analysis reveals an exceptional predictive fit. The adjusted coefficient of determination (adjusted $r^{2}$) of 0.99919 is extraordinarily high, indicating that 99.92% of the variability in calculated solubility is explained by experimental solubility. This value, which is close to 1, suggests that there is almost perfect agreement between the experimental data and the values predicted by the theoretical model.

Analysis of variance (ANOVA), on the other hand, confirms the statistical significance of the regression model as a whole. The *F*-statistic is 232,934.22, and the *p*-value is 2.06$\cdot 10^{-291}$. Thus, the null hypothesis that the experimental variable has no effect on the calculated variable is rejected. This affirms that the model is statistically robust and not a product of chance.

In general, the Yalkowsky–Roseman–Van’t Hoff model is an excellent predictor of SMT solubility within the analyzed data range. There is a positive, strong, and statistically significant linear relationship with an almost perfect 1:1 correspondence between the calculated and experimental values.

## 4. Materials and Methods

### 4.1. Reagents

For developing the research, the following analytical grade reagents were used: sulfamethazine (Sigma-Aldrich, Burlington, MA, USA), ethanol (Sigma-Aldrich, Burlington, MA, USA), acetonitrile (Supelco, Burlington, MA, USA), NaOH (Sigma-Aldrich, Burlington, MA, USA), and the double-distilled water (component 2) with conductivity lower than 2 us/cm.

### 4.2. Solubility Determination

Solubility determination was carried out using the shake flask method, following the methodology proposed by Higuchi and Connors [66,67]. The procedure involves preparing saturated solutions in amber-colored flasks by adding an excess of the drug (SMT) to a defined volume of pure solvent (MeCN or EtOH) or cosolvent mixture {MeCN (1) + EtOH (2)} while vigorously shaking. Nineteen cosolvent mixtures {MeCN (1) + EtOH (2)} were prepared gravimetrically, using a four-decimal place analytical balance with a readability of ±0.0001 g (RADWAG AS 220.R2, Warsaw, Poland) [26]. To ensure thermodynamic equilibrium, the samples were kept at a constant temperature in a recirculating bath for 36 h, allowing for the coexistence of a saturated solution and an undissolved solid phase. To ensure sample saturation, the concentration of the solution was periodically measured until it reached a constant value. Once equilibrium was reached, drug quantification was performed by taking a sample of the saturated solution. To remove any undissolved solids before quantification, the samples were filtered through 0.45 µm membranes (Millipore Corp. Swinnex-13, Atlanta, GA, USA) and diluted with a 0.1 N NaOH solution. We quantified the drug concentration by UV-Vis spectrometry (UV/Vis EMC-11- UV spectrophotometer, Duisburgo, Germany) after determining the wavelength of maximum absorbance (268 nm) and using a calibration curve that complies with the Lambert–Beer law. Finally, differential scanning calorimetry (DSC) was used to analyze the solid phase in equilibrium with the saturated solution to evaluate possible polymorphic changes or decomposition of the drug during the experiment. The solid phase in equilibrium was obtained by saturating the sample at a temperature higher than the equilibrium temperature to obtain a precipitate at the bottom of each flask.

### 4.3. Calorimetric Study

The temperature and enthalpy of melting of four SMT samples were determined using differential scanning calorimetry (DSC 204 F1; Phoenix, Dresden, Germany). The equipment was pre-calibrated using indium and tin as standards, and an empty, sealed capsule was used as a reference. Approximately 10.0 mg of each sample was placed in an aluminum crucible and introduced into the calorimeter under a nitrogen flow of 10 mL/min. A heating cycle from 380 to 500 K with a ramp rate of 10 K/min was then applied. Prior to analysis, the solid samples in equilibrium with the saturated solution were dried at room temperature for 48 h under a continuous stream of dry air [26].

## 5. Conclusions

The solubility of SMT in cosolvent mixtures of MeCN and EtOH is primarily determined by the energetics of molecular interactions rather than the solubility parameter of the cosolvent mixtures. Solubility increases almost linearly with the proportion of MeCN because MeCN is a considerably more efficient solvent for SMT than EtOH. The dissolution process is endothermic and enthalpy-driven over the entire composition range, meaning the main energy barrier is the breakdown of cohesive interactions. Specifically, the energetic cost of forming a cavity in the EtOH hydrogen bond network limits solubility in EtOH-rich mixtures. Adding MeCN, a solvent with weaker intermolecular interactions, reduces this enthalpic cost and favors the dissolution of SMT, in addition to increasing ($\Delta_{\mathrm{mix}}S^{\circ }$). The enthalpy–entropy compensation phenomenon explains these observations, showing that the solubility of SMT results from a balance between the cost of cavity formation and specific solute–solvent interactions. Together with confirmation that the drug does not exhibit polymorphism upon recrystallization, these findings provide a robust physicochemical basis and crucial data for designing and optimizing pharmaceutical processes, such as the crystallization and formulation of sulfamethazine. Finally, the van’t Hoff–Yalkowsky–Roseman model demonstrated an “almost perfect” correlation with the experimental data. It presented a low average relative deviation and a high coefficient of determination. This shows that the model is statistically robust and a valuable tool for correlating the solubility of SMT in the studied cosolvent system.

## Figures and Tables

**Figure 1 molecules-30-03590-f001:**
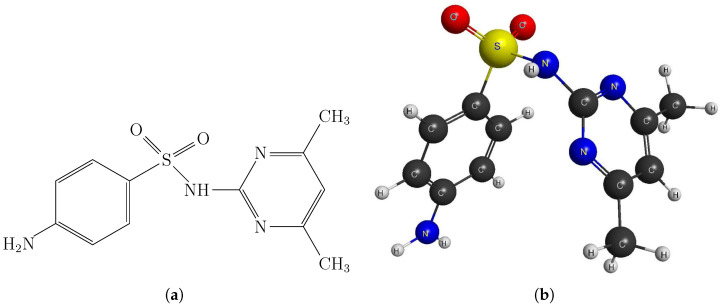
(**a**) The 2D and (**b**) 3D chemical molecular structure of SMT (molecular formula C_12_H_14_N_4_O_2_S; IUPAC name: 4-amino-*N*-(4,6-dimethylpyrimidin-2-yl)benzenesulfonamide; CAS: 57-68-1) [20].

**Figure 2 molecules-30-03590-f002:**
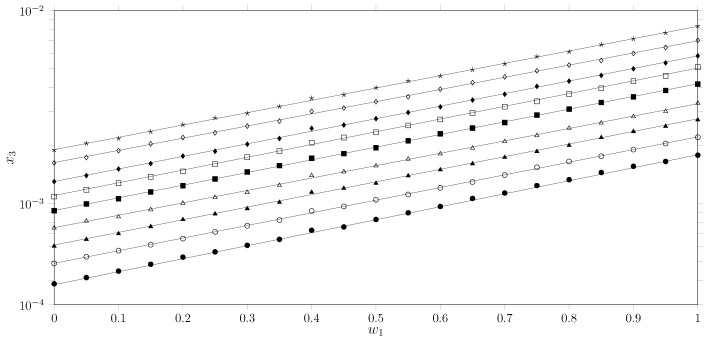
Mole fraction of SMT depending on the mass fraction of MeCN in the {MeCN (1) + EtOH (2)} mixtures free of SMT. •: 278.15 K; ∘: 283.15 K; ▴: 288.15 K; ▵: 293.15 K; ▪: 298.15 K; □: 303.15 K; ⧫: 308.15; ⋄: 313.15 and ★: 318.15 K.

**Figure 3 molecules-30-03590-f003:**
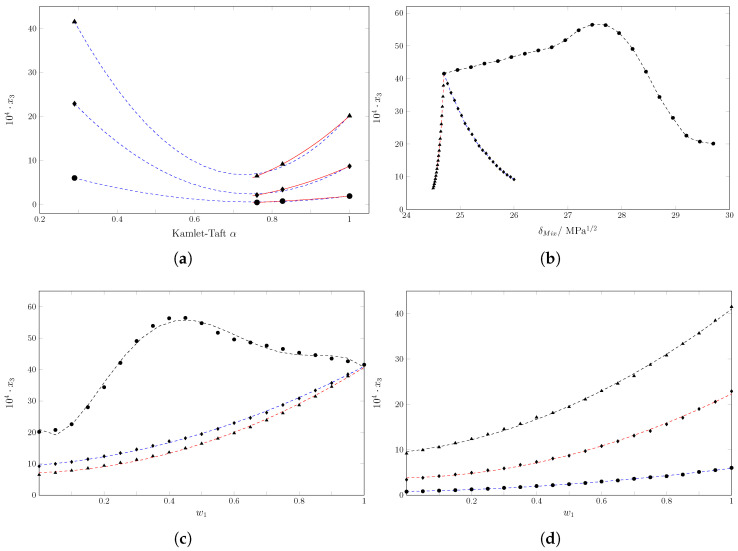
Trends in the solubility behavior of SD, SMR, and SMT at 298.15 K: (**a**) solubility of SD (•), SMR (⧫) and SMT (▴) as a function of Kamlet–Taft acidity parameters (α); (**b**) solubility of SD (•), SMR (⧫) and SMT (▴) as a function of the solubility parameter of the cosolvent mixture; (**c**) solubility of SMT in cosolvent mixtures: MeCN + MeOH (•), MeCN + EtOH (⧫), MeCN + PrOH (▴); (**d**) solubility of SD (•), SMR (⧫) and SMT (▴) in MeCN + EtOH cosolvent mixtures.

**Figure 4 molecules-30-03590-f004:**
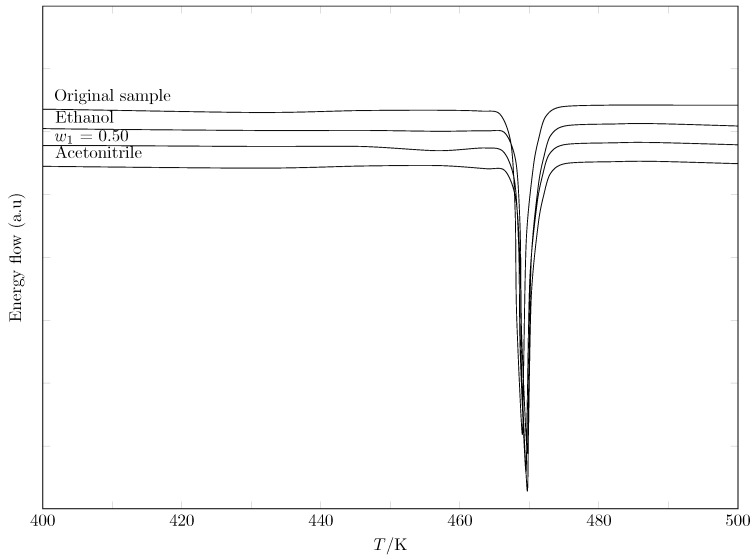
DSC curves of different SMT samples.

**Figure 6 molecules-30-03590-f006:**
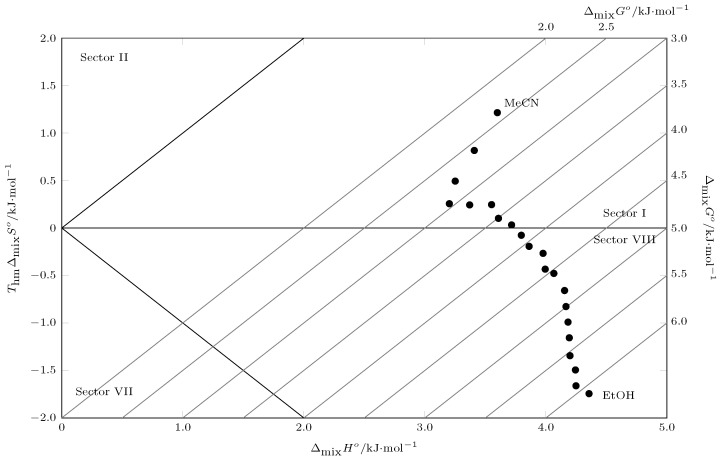
Perlovich’s plot for the mixing process of SMT (3) in cosolvent mixtures {MeCN (1) + 1 − EtOH (2)} at $T_{\mathrm{hm}}$/K = 297.6. The gray lines represent the Gibbs energy of mixing.

**Figure 7 molecules-30-03590-f007:**
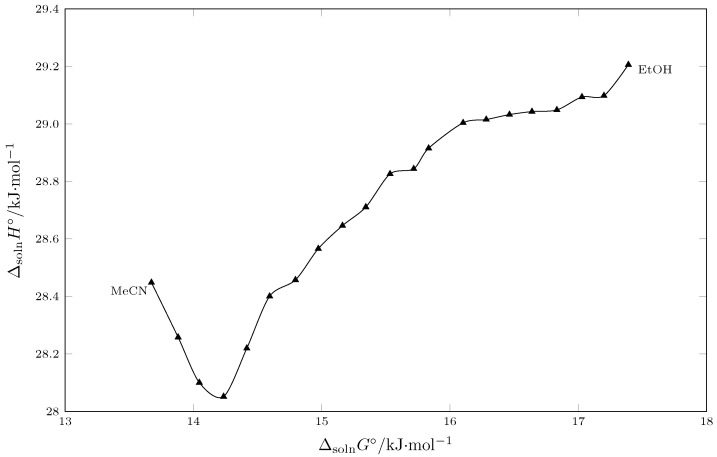
Enthalpic–entropic compensation corresponding to the solution process of the SMT in cosolvent mixtures {MeCN (1) + EtOH (2)} at $T_{\mathrm{hm}}$/K = 297.6.

**Figure 8 molecules-30-03590-f008:**
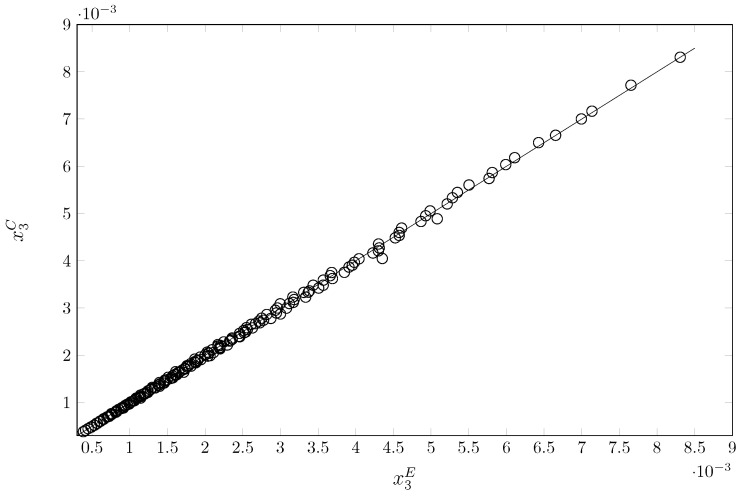
Calculated solubility vs. experimental solubility of SMT in two pure solvents (MeCN and EtOH) and 19 MeCN + EtOH cosolvent mixtures at nine temperatures (between 278.15 and 318.15 K).

**Table 3 molecules-30-03590-t003:** Apparent thermodynamic functions relative to solution processes of SMT (3) in MeCN (1) + EtOH (2) mixtures at $T_{\mathrm{hm}}$/K =297.6 as a function the mass fraction of MeCN (1) ($w_{1}$) in the {MeCN (1) + EtOH (2)} mixtures free of SMT (3) ^a^.

*w*_1_ ^b^	$\Delta_{\mathrm{soln}}G^{\circ }$	$\Delta_{\mathrm{soln}}H^{\circ }$	$\Delta_{\mathrm{soln}}S^{\circ }$	$T_{\mathrm{hm}}\Delta_{\mathrm{soln}}S^{\circ }$	$\zeta_{H}$ ^c^	$\zeta_{\mathrm{TS}}$ ^c^
	/kJ·${\mathrm{mol}}^{-1}$	/kJ·${\mathrm{mol}}^{-1}$	/J·$K^{-1}$·${\mathrm{mol}}^{-1}$	/kJ·${\mathrm{mol}}^{-1}$		
0.00	17.39 ± 0.16	29.21 ± 0.27	39.7 ± 1.7	11.82 ± 0.31	0.712	0.288
0.05	17.2 ± 0.16	29.1 ± 0.29	40.0 ± 2.0	11.9 ± 0.3	0.710	0.290
0.10	17.03 ± 0.16	29.09 ± 0.24	40.5 ± 1.5	12.06 ± 0.29	0.707	0.293
0.15	16.83 ± 0.2	29.05 ± 0.3	41.0 ± 2.3	12.2 ± 0.4	0.704	0.296
0.20	16.64 ± 0.15	29.04 ± 0.25	41.7 ± 1.5	12.4 ± 0.29	0.701	0.299
0.25	16.46 ± 0.18	29.03 ± 0.3	42.2 ± 2.1	12.6 ± 0.3	0.698	0.302
0.30	16.28 ± 0.17	29.02 ± 0.31	42.8 ± 2.2	12.7 ± 0.4	0.695	0.305
0.35	16.10 ± 0.17	29.00 ± 0.31	43.4 ± 2.2	12.9 ± 0.4	0.692	0.308
0.40	15.83 ± 0.16	28.92 ± 0.18	44.0 ± 1.0	13.08 ± 0.24	0.689	0.311
0.45	15.72 ± 0.16	28.84 ± 0.28	44.1 ± 1.9	13.13 ± 0.33	0.687	0.313
0.50	15.53 ± 0.18	28.83 ± 0.19	44.7 ± 1.2	13.29 ± 0.26	0.684	0.316
0.55	15.34 ± 0.16	28.71 ± 0.24	44.9 ± 1.4	13.37 ± 0.29	0.682	0.318
0.60	15.16 ± 0.15	28.65 ± 0.31	45.3 ± 2.1	13.5 ± 0.3	0.680	0.320
0.65	14.97 ± 0.19	28.57 ± 0.31	45.7 ± 2.3	13.6 ± 0.4	0.678	0.322
0.70	14.80 ± 0.18	28.46 ± 0.25	45.9 ± 1.7	13.66 ± 0.31	0.676	0.324
0.75	14.59 ± 0.13	28.4 ± 0.28	46.4 ± 1.7	13.81 ± 0.31	0.673	0.327
0.80	14.42 ± 0.16	28.22 ± 0.26	46.4 ± 1.6	13.8 ± 0.3	0.672	0.328
0.85	14.24 ± 0.16	28.05 ± 0.33	46.4 ± 2.4	13.8 ± 0.4	0.670	0.330
0.90	14.05 ± 0.16	28.1 ± 0.29	47.2 ± 1.9	14.05 ± 0.33	0.667	0.333
0.95	13.88 ± 0.11	28.26 ± 0.31	48.3 ± 1.9	14.38 ± 0.33	0.663	0.337
1.00	13.66 ± 0.16	28.5 ± 0.6	50 ± 6	14.8 ± 0.6	0.658	0.342

^a^ ± are the standard deviations, ^b^
$w_{1}$ is the mass fraction of MeCN (1) in the {MeCN (1) + EtOH (2)} mixtures free of SMT (3). ^c^
$\zeta_{H}$ and $\zeta_{TS}$ are the relative contributions by enthalpy and entropy toward Gibbs energy of solution. These values were calculated by means of Equations (Equation 4) and (Equation 5).

**Table 4 molecules-30-03590-t004:** Apparent thermodynamic functions relative to the mixing processes of SMT (3) in {MeCN (1) + EtOH (2)} mixtures at $T_{\mathrm{hm}}$/K = 297.6 as a function the mass fraction of MeCN (1) ($w_{1}$) in the {MeCN (1) + EtOH (2)} mixtures free of SMT (3) ^a^.

$w_{1}$ ^b^	$\Delta_{\mathrm{mix}}G^{\circ }$	$\Delta_{\mathrm{mix}}H^{\circ }$	$\Delta_{\mathrm{mix}}S^{\circ }$	$T_{\mathrm{hm}}\Delta_{\mathrm{mix}}S^{\circ }$
	/kJ·${\mathrm{mol}}^{-1}$	/kJ·${\mathrm{mol}}^{-1}$	/J·${\mathrm{mol}}^{-1}\cdot$$K^{-1}$	/kJ·${\mathrm{mol}}^{-1}$
0.00	6.1 ± 0.19	4.4 ± 0.4	−5.9 ± 2.0	−1.74 ± 0.33
0.05	5.91 ± 0.19	4.2 ± 0.4	−5.6 ± 2.2	−1.7 ± 0.4
0.10	5.74 ± 0.19	4.2 ± 0.4	−5.0 ± 1.8	−1.49 ± 0.31
0.15	5.54 ± 0.22	4.2 ± 0.4	−4.5 ± 2.5	−1.3 ± 0.4
0.20	5.35 ± 0.18	4.2 ± 0.4	−3.9 ± 1.8	−1.16 ± 0.31
0.25	5.17 ± 0.2	4.2 ± 0.4	−3.3 ± 2.3	−1.0 ± 0.4
0.30	4.99 ± 0.20	4.2 ± 0.4	−2.8 ± 2.4	−0.8 ± 0.4
0.35	4.81 ± 0.20	4.2 ± 0.4	−2.2 ± 2.4	−0.7 ± 0.4
0.40	4.54 ± 0.18	4.1 ± 0.4	−1.6 ± 1.4	−0.48 ± 0.26
0.45	4.43 ± 0.19	4.0 ± 0.4	−1.5 ± 2.1	−0.4 ± 0.3
0.50	4.24 ± 0.2	4.0 ± 0.4	−0.9 ± 1.6	−0.27 ± 0.28
0.55	4.05 ± 0.19	3.9 ± 0.4	−0.6 ± 1.7	−0.19 ± 0.3
0.60	3.87 ± 0.18	3.8 ± 0.4	−0.3 ± 2.3	−0.1 ± 0.4
0.65	3.68 ± 0.21	3.7 ± 0.4	0.1 ± 2.5	0.0 ± 0.4
0.70	3.51 ± 0.20	3.6 ± 0.4	0.3 ± 1.9	0.10 ± 0.32
0.75	3.30 ± 0.16	3.6 ± 0.4	0.8 ± 1.9	0.25 ± 0.33
0.80	3.13 ± 0.19	3.4 ± 0.4	0.8 ± 1.9	0.24 ± 0.32
0.85	2.95 ± 0.19	3.2 ± 0.4	0.9 ± 2.6	0.3 ± 0.4
0.90	2.76 ± 0.19	3.3 ± 0.4	1.7 ± 2.1	0.5 ± 0.3
0.95	2.59 ± 0.15	3.4 ± 0.4	2.7 ± 2.2	0.8 ± 0.3
1.00	2.37 ± 0.19	3.6 ± 0.6	4 ± 6	1.2 ± 0.6

^a^ (±) are the standard deviations, ^b^
$w_{1}$ is the mass fraction of MeCN (1) in the {MeCN (1) + EtOH (2)} mixtures free of SMT (3).

## Data Availability

Data are contained within the article.

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
