# Peer review of "Solubility of Sulfamethazine in Acetonitrile–Ethanol Cosolvent Mixtures: Thermodynamic Analysis and Mathematical Modeling"

_molecules, 2025, doi:10.3390/molecules30173590_

Round 1

Reviewer 1 Report

Comments and Suggestions for Authors

Caviedes-Rubio et al. present a study of the solubility of Sulfamethazine in a mixture of two solvents (cosolvent), acetonitrile and ethanol. They performed experimental studies, analyzed the data with robust models and also used a predictive tool for further validation of their results. The manuscript is presented in a readable manner, easy to follow, the results are well presented, and the conclusions are in line with the data presented.

I present to the authors my concerns.

A few issues of concern are:

  1. In the methods section the authors mention that they prepare saturated solutions. My question is, how do you know a priori, what would be a proper saturated solution?
  2. The arguments presented for the use of ethanol are not as clear as they should be. In the manuscript it is shown that methanol has a better (fig 3), albeit not linear, solubility when used in combination with acetonitrile. Perhaps the authors should lay out their previous work on SMT and other similar antibiotics in a clearer manner, this would help the reader understand the background of the present work.
  3. The authors should discuss the possible limitations of the work. For example: Is the use of these solvent viable for the pharmaceutical industry from an economic and green chemistry perspective?

Minor details are:

  • Lines 52 to 53 are in Spanish. Correct this please.
  • Line 63: There is a “00g)” that does not seem to fit the context.
  • Figure 3: In the caption for figure 3d both SMR and SMT have the same symbol.
  • Line 159: Elaborate more on the link with solid state energetics.
  • Line 167: I think the reference should be Figure 4 and not 3.
  • Line 195: I think the reference should be Table 3 and not 2.
  • Line 218-219. Please explain in more detail the different sectors in a Perlovich diagram and the implications. This will help the reader understand the results better.
  • Line 240: What do you mean by “the thermodynamic quantities of the SMT melt”? I do not quite understand this phrase.
  • Table 4 and Figure 7 are not reference in the text (if you attend to a previous comment this might change for Table 4).
  • Captions for Tables 3 and 4 are no really clear, particularly for Table 4 “Apparent thermodynamic functions relative to of mixing processes of SMT”. The term “relative to […] processes” is not really clear to me.
  • In figures 5 and 6 you use {MeCN (1) + 1-EtOH (2)}, could you please clarify this change in cosolvent nomenclature as it is not mentioned in the text?
  • In lines 291 and 292 I do not understand the sentence “This makes the sulfamethazine solution the most favorable in terms of free energy”. Which SMT solution? At what w1 value? It seems there is some information lacking.
  • In equation 9 please define the i index.
  • The authors could explain the goodness of fit in a clearer way. How it is presented now seems a bit convoluted.
  • Finally, mention the computational validation in the conclusions.
Comments on the Quality of English Language

In general, the quality of English is good. However, clarity needs to be improved throughout the manuscript.

Author Response

Please find attached the corrected version of the manuscript, "Solubility of Sulfamethazine in Acetonitrile-Ethanol Cosolvent Mixtures: Thermodynamic Analysis and Mathematical Modeling" (Manuscript ID: molecules-3788649). First, we would like to thank you for your helpful and insightful suggestions. This new version has been organized according to the reviewers' suggestions. These changes have allowed us to improve our work. All changes have been highlighted in yellow in the new version.

Reviewer 2 Report

Comments and Suggestions for Authors

The manuscript appears to be a continuation of a series of publications by the authors with a rather similar scope. Nevertheless, the data presented seem reliable, and the discussion is convincing, warranting publication. However, there are several issues that should be addressed:

  • Abstract: “Thermodynamic analysis revealed that dissolution is a non-spontaneous, enthalpy-driven process for all compositions.” Non-spontaneous process would not happen. Dissolution is non-spontaneous at the standard-state conditions (see also later). How is an endothermic process enthalpy-driven?
  • In the Introduction, it should be stated why particularly ethanol and acetonitrile have been selected among the other cosolvents. Why not use EtOH for ethanol in this section and the Abstract, if MeCN stands for acetonitrile.
  • L32 Solvents do not possess a gap in literature.
  • L52−53 Spanish
  • L61 - missing space between mixture and {MeCN
  • L63 - 00 g?
  • L77 - unnecessary space before the last period
  • L79 - enthalpy of fusion (as used later in the text)
  • L89−91 - listing the temperatures is unnecessary as their values are given in Table 1
  • L104 pKa1 - p should be in regular font, K in italics, a1 in regular and subscript. “Labile proton” sounds odd.
  • Table 2 “enthalpy of fusion” instead of “enthalpy” in the Caption. It is unnecessary to write enthalpy of fusion and melting point again in the Table, since the notation is clear.
  • Figure 4 in the Caption, it should be stated DSC curves (or thermograms). On the y-axis, A.U. instead of A.U.
  • Eq. (2): intercept?
  • Please check Eq. (4).
  • L201−202 - the sentence “The solution enthalpy (∆solnH°) is positive across the entire range of compositions, confirming that the process is endothermic.” is unnecessary. The same holds for the corresponding sentence in Line 250.
  • L205 EtOH lattice?
  • L206 The difference “from 29.21 to 28.5 kJ mol−1” is often within the experimental error range; the statements that follow and revolve around the small (-0.7 kJ mol−1) difference should be attenuated.

L210−212

  • Kelvin (K) should not be in italics, and the “positive entropy” is not correct phrasing as this corresponds to the change in entropy.
  • The sentence “A positive entropy value is thermodynamically favorable and reflects the increased probability of interactions in the system when a crystalline solid dissolves.” seems odd - how can entropy increase if the “probability of interactions” increases? It is possible, but then other effects should be taken into account. The authors should focus on the dispersion of energy (interactions) in crystal vs. in the solution (where the dispersion is obviously more pronounced).
  • Figure 5: Gibbs energy of solution
  • Figures 5–7: kJ∙mol−1 instead of kJ.mol−1
  • L235 studied temperature instead of study temperature
  • L241 Thm / K = 297.6 K
  • L244−246 “The Gibbs energy of mixing (∆mixG◦) is positive across the entire range of compositions, indicating that the process of integrating the SMT molecule into the solvent is not favorable.” This value corresponds to the standard Gibbs energy of mixing. The authors should take into account the difference between ΔmixG° and ΔmixG when discussing the quantities.
  • L256 “system’s physical chemistry” sounds odd and vague.
  • L257 “entropy” cannot be negative - the change in entropy is. Mistakes of this kind can be found throughout the manuscript and must be corrected.
  • L302−303 A, B, and T should not be in bold.
  • L303 kelvin instead of Kelvin (throughout the manuscript)

Abbreviations

  • standard reaction thermodynamic quantities where appropriate (ΔsolnX°, ΔmixX°)
  • J and K are physical units and should be removed from the list.

In all tables it should be stated how the errors of the listed values were assessed.

Author Response

  1. Abstract: “Thermodynamic analysis revealed that dissolution is a non-spontaneous, enthalpy-driven process for all compositions.” Non-spontaneous process would not happen. Dissolution is non-spontaneous at the standard-state conditions (see also later). How is an endothermic process enthalpy-driven?. Response: We completely agree. The statement has been corrected.
  2. In the Introduction, it should be stated why particularly ethanol and acetonitrile have been selected among the other cosolvents. Why not use EtOH for ethanol in this section and the Abstract, if MeCN stands for acetonitrile.Response: Lines 29-43: A clearer justification is provided for the use of solvents.
  3. L32 Solvents do not possess a gap in literature. Response: Lines 29-43: A clearer justification is provided for the use of solvents.
  4. L52−53 Spanish: Response: The phrase was translated into English.
  5. L61 - missing space between mixture and {MeCN Response: The observation has been corrected.
  6. L63 - 00 g?: Response: Typo fixed.
  7. L77 - unnecessary space before the last period Response: The observation has been corrected.
  8. L79 - enthalpy of fusion (as used later in the text). Response: The observation has been corrected.
  9. L89−91 - listing the temperatures is unnecessary as their values are given in Table 1 Response: The observation has been corrected.
  10. L104 pKa1 - p should be in regular font, K in italics, a1 in regular and subscript. “Labile proton” sounds odd. Response: The observation has been corrected.
  11. Table 2 “enthalpy of fusion” instead of “enthalpy” in the Caption. It is unnecessary to write enthalpy of fusion and melting point again in the Table, since the notation is clear. Response: The observation has been corrected.
  12. Figure 4 in the Caption, it should be stated DSC curves (or thermograms). On the y-axis, A.U. instead of A.U. Response: The observation has been corrected.
  13. Eq. (2): intercept? Response: The word "intercept" was changed to "a" to make the equation clearer.
  14. Please check Eq. (4). Response: The equation has been corrected.
  15. L201−202 - the sentence “The solution enthalpy (∆solnH°) is positive across the entire range of compositions, confirming that the process is endothermic.” is unnecessary. The same holds for the corresponding sentence in Line 250. Response: The observation has been corrected.
  16. L205 EtOH lattice? Response: The wording was changed for clarity.
  17. L206 The difference “from 29.21 to 28.5 kJ mol−1” is often within the experimental error range; the statements that follow and revolve around the small (-0.7 kJ mol−1) difference should be attenuated. Response: Thank you very much for your observation. We agree completely. The comment has been rewritten in more appropriate terms.
  18. L210−212: Kelvin (K) should not be in italics, and the “positive entropy” is not correct phrasing as this corresponds to the change in entropy. Response: The observation has been corrected.
  19. The sentence “A positive entropy value is thermodynamically favorable and reflects the increased probability of interactions in the system when a crystalline solid dissolves.” seems odd - how can entropy increase if the “probability of interactions” increases? It is possible, but then other effects should be taken into account. The authors should focus on the dispersion of energy (interactions) in crystal vs. in the solution (where the dispersion is obviously more pronounced). Response: The sentence was rewritten for clarity.
  20. Figure 5: Gibbs energy of solution. Response: The observation has been corrected
  21. Figures 5–7: kJ∙mol−1 instead of kJ.mol−1 Response: The observation has been corrected
  22. L235 studied temperature instead of study temperature. Response: The observation has been corrected
  23. L241 Thm / K = 297.6 K. Response: The observation has been corrected
  24. L244−246 “The Gibbs energy of mixing (∆mixG◦) is positive across the entire range of compositions, indicating that the process of integrating the SMT molecule into the solvent is not favorable.” This value corresponds to the standard Gibbs energy of mixing. The authors should take into account the difference between ΔmixG° and ΔmixG when discussing the quantities. Response: The sentence was rewritten for clarity.
  25. L256 “system’s physical chemistry” sounds odd and vague. Response: Thank you for these comments. They have greatly improved the manuscript.
  26. L257 “entropy” cannot be negative - the change in entropy is. Mistakes of this kind can be found throughout the manuscript and must be corrected. Response: The observation has been corrected
  27. L302−303 A, B, and T should not be in bold. Response: The observation has been corrected
  28. L303 kelvin instead of Kelvin (throughout the manuscript. Response: The observation has been corrected
  29. Abbreviations: standard reaction thermodynamic quantities where appropriate (ΔsolnX°, ΔmixX°), J and K are physical units and should be removed from the list. Response:  The observation has been corrected
  30. In all tables it should be stated how the errors of the listed values were assessed. Response: The observation has been corrected

Reviewer 3 Report

Comments and Suggestions for Authors

The manuscript deals with the solubility behavior of sulfamethazine in acetonitrile-ethanol
mixed-solvents over the entire range of solvent composition within a temperature spanned
around ambient conditions, and follows a similar strategy pursued previously in a series of
solubility studies by the leading authors. While the manuscript is well written, with detailed
tabulation of results, including their interpretation, I have a few observations and suggestions
to strengthen the impact of this work as described below:
1) The subtitle indicates “predictive study”, yet, what the authors deliver from this modeling
effort is a pretty good “correlative” rather than “predictive” tool.
2) In lines 103-108, the authors declare that “The main acidic site is the sulfonamide proton,
which has an acid dissociation constant (pKa1) of 7.4 [25,26]. This proton is labile and is the
main site of interaction with basic solvents. The main basic site is the amino group of aniline,
with an acid dissociation constant, pKa2, of 2.65 for its conjugate acid. However, this site is
protonated in strongly acidic media; therefore, in neutral solvents, SMT is likely to be
unprotonated” As far I understand, these data are for aqueous solutions; what about the
actual acid/base behavior of SMT in the acetonitrile-ethanol mixed-solvent environment? In
addition, what are you referring to as “strongly acidic media”? Note that in line 139 the
authors claim “SMT is an acid (sulfonamide proton) that seeks a basic environment for
favorable interaction.” So, what is the actual pKa of SMT in these non-aqueous
environments? Please, discuss and clarify by addressing the identified issues.
3) In the title of Table 1, it appears to have a missing 10000-prefactor for the quoted solubility
data when compared with the results in Figure 2.
4) In lines 253-254 the authors mention “As MeCN is added, a solvent with weaker
intermolecular interactions (dipole-dipole), the cost of forming such a cavity decreases. This
is reflected in the reduction of ∆mixH◦.” This is an interesting, yet tricky, situation since the
contrasting dipole moments and polarizabilities of the solvent and cosolvent, i.e., 3.9 D and
3.4Å^3 for acetonitrile and 1.7 D and 1.4Å^3 for ethanol, while ~12D and ~28Å^3 for SMT.
Therefore, the dipole-dipole rationale to interpret the results might not be an accurate view
given the obvious and significant polarization contributions. By the way, what is meant by “
a solvent with weaker intermolecular interactions (dipole-dipole)” …weaker than that of the
ethanol… dipole-induced dipole interactions…?
5) In lines 259-264 the authors say “Negative mixing entropy clearly indicates the ordering or
structuring of the solvent around the solute molecule. In the protic environment of EtOH,
SMT organizes EtOH molecules to maximize hydrogen bonding interactions, resulting in a
net decrease in system disorder. In contrast, positive entropy in the MeCN environment
indicates a more conventional mixing process where introducing the solute increases overall
randomness.” I find these statements rather vague and conjectural for the following
reasons:
a) The claims above presume some (already) established link between the magnitude of
the mixing entropy and some structuring/ordering of the liquid environments around
the SMT. Not aware of such link. Please discuss/clarify,
b) Structuring/ordering means little to nothing unless there is a clear definition of it,
c) More conventional mixing process without a clear definition sounds as if there is a “less
conventional mixing process”. Neither one conveys a definite description of the actual
mixing process. Please clarify by describing/differentiating/identifying the “two” types
of processes.
d) The solute increases overall randomness does not add anything but confusion.
Randomness does not belong to the language of solvation thermodynamics…
Thus, revise all the statements accordingly.
6) In lines 337-338 the authors indicate “The solubility of SMT in cosolvent mixtures of MeCN
and EtOH is primarily determined by the energetics of molecular interactions rather than the
polarity of the medium.” Given the magnitude of the species dipoles and polarizabilities, I
would assume that the alleged energetic would be determined by the polarity/polarizability
of the medium interacting with those corresponding to the SMT. Thus, where is the catch?
7) In lines 343=344, the authors claim “Adding MeCN, a solvent with weaker intermolecular
interactions, reduces this enthalpic cost and favors dissolution.” Weaker intermolecular
interactions than what/which one? Please clarify.
8) In lines 345-347, the authors write “This behavior is corroborated by a negative entropy of
mixing in EtOH-rich environments, which is indicative of an ordering of the solvent molecules
around the solute.” Suffers the same fate as item 5.a above, therefore, revise/justify.
Finally, please cite the original work/s when referring to the pKa of SMT as Ref. 25. In fact, they
are, instead:
• Park, J. Y.; Huwe, B., Effect of H and soil structure on transport of sulfonamide
antibiotics in agricultural soils. Int. J. Environ. Pollut. 2016, 213, 561-570, DOI:
10.1016/j.envpol.2016.01.089.
• Kang, S. I.; Bae, Y . H., Ph-induced solubility transition of sulfonamide-based polymers. J.
CR 2002, 80, 145-155, DOI: 10.1016/S0168-3659(02)00021-4.
Please correct the format of several citations in the reference section, i.e. there is a duplication
in the DOI portion as “https://doi.org/https://doi.org/...”

Author Response

  1. The subtitle indicates “predictive study”, yet, what the authors deliver from this modeling
    effort is a pretty good “correlative” rather than “predictive” tool. Response: We appreciate the comment, we agree, so we have decided to change the title.
  2. In lines 103-108, the authors declare that “The main acidic site is the sulfonamide proton,
    which has an acid dissociation constant (pKa1) of 7.4 [25,26]. This proton is labile and is the
    main site of interaction with basic solvents. The main basic site is the amino group of aniline,
    with an acid dissociation constant, pKa2, of 2.65 for its conjugate acid. However, this site is
    protonated in strongly acidic media; therefore, in neutral solvents, SMT is likely to be
    unprotonated” As far I understand, these data are for aqueous solutions; what about the
    actual acid/base behavior of SMT in the acetonitrile-ethanol mixed-solvent environment? In
    addition, what are you referring to as “strongly acidic media”? Note that in line 139 the
    authors claim “SMT is an acid (sulfonamide proton) that seeks a basic environment for
    favorable interaction.” So, what is the actual pKa of SMT in these non-aqueous
    environments? Please, discuss and clarify by addressing the identified issues. Response: After analyzing the evaluator's suggestions, we have decided to remove this discussion entirely. In addition to being irrelevant, the analysis may be incorrect. We greatly appreciate the evaluator's observation.
  3. In the title of Table 1, it appears to have a missing 10000-prefactor for the quoted solubility
    data when compared with the results in Figure 2. Response: The observation was taken into account, and the factor of 10⁴ was indicated.
  4. In lines 253-254 the authors mention “As MeCN is added, a solvent with weaker
    intermolecular interactions (dipole-dipole), the cost of forming such a cavity decreases. This
    is reflected in the reduction of ∆mixH◦.” This is an interesting, yet tricky, situation since the
    contrasting dipole moments and polarizabilities of the solvent and cosolvent, i.e., 3.9 D and
    3.4Å^3 for acetonitrile and 1.7 D and 1.4Å^3 for ethanol, while ~12D and ~28Å^3 for SMT.
    Therefore, the dipole-dipole rationale to interpret the results might not be an accurate view
    given the obvious and significant polarization contributions. By the way, what is meant by “
    a solvent with weaker intermolecular interactions (dipole-dipole)” …weaker than that of the
    ethanol… dipole-induced dipole interactions…? Response: The observation was taken into account, and the factor of 10⁴ was indicated.
  5. In lines 259-264 the authors say “Negative mixing entropy clearly indicates the ordering or
    structuring of the solvent around the solute molecule. In the protic environment of EtOH,
    SMT organizes EtOH molecules to maximize hydrogen bonding interactions, resulting in a
    net decrease in system disorder. In contrast, positive entropy in the MeCN environment
    indicates a more conventional mixing process where introducing the solute increases overall
    randomness.” I find these statements rather vague and conjectural for the following
    reasons:
    a) The claims above presume some (already) established link between the magnitude of
    the mixing entropy and some structuring/ordering of the liquid environments around
    the SMT. Not aware of such link. Please discuss/clarify,
    b) Structuring/ordering means little to nothing unless there is a clear definition of it,
    c) More conventional mixing process without a clear definition sounds as if there is a “less
    conventional mixing process”. Neither one conveys a definite description of the actual
    mixing process. Please clarify by describing/differentiating/identifying the “two” types
    of processes.
    d) The solute increases overall randomness does not add anything but confusion.
    Randomness does not belong to the language of solvation thermodynamics…
    Thus, revise all the statements accordingly. Response: Once again, we appreciate the reviewer's pertinent comments, which prevented us from using ambiguous wording. In this context, we reevaluated the discussion and presented a more rigorous thermodynamic analysis.
  6. In lines 337-338 the authors indicate “The solubility of SMT in cosolvent mixtures of MeCN
    and EtOH is primarily determined by the energetics of molecular interactions rather than the
    polarity of the medium.” Given the magnitude of the species dipoles and polarizabilities, I
    would assume that the alleged energetic would be determined by the polarity/polarizability
    of the medium interacting with those corresponding to the SMT. Thus, where is the catch? Response: We changed the term "polarity" to "solubility parameter," which we had mistakenly used as a synonym.
  7.  In lines 343=344, the authors claim “Adding MeCN, a solvent with weaker intermolecular
    interactions, reduces this enthalpic cost and favors dissolution.” Weaker intermolecular
    interactions than what/which one? Please clarify. Response: The paragraph was rewritten to make it clearer for the reader.
  8. In lines 345-347, the authors write “This behavior is corroborated by a negative entropy of
    mixing in EtOH-rich environments, which is indicative of an ordering of the solvent molecules
    around the solute.” Suffers the same fate as item 5.a above, therefore, revise/justify. Response: The observation was addressed by changing the wording of the paragraph.
    Finally, please cite the original work/s when referring to the pKa of SMT as Ref. 25. In fact, they
    are, instead:
    • Park, J. Y.; Huwe, B., Effect of H and soil structure on transport of sulfonamide
    antibiotics in agricultural soils. Int. J. Environ. Pollut. 2016, 213, 561-570, DOI:
    10.1016/j.envpol.2016.01.089.
    • Kang, S. I.; Bae, Y . H., Ph-induced solubility transition of sulfonamide-based polymers. J.
    CR 2002, 80, 145-155, DOI: 10.1016/S0168-3659(02)00021-4.
    Please correct the format of several citations in the reference section, i.e. there is a duplication
    in the DOI portion as “https://doi.org/https://doi.org/...” 

Round 2

Reviewer 3 Report

Comments and Suggestions for Authors

As I mentioned previously, it is highly desirable practice to cite original pieces of works rather than referring indirectly to them. Therefore, the authors should cite the proper ones when referring to the pKa of SMT as Ref. 25., i.e., 

• Park, J. Y.; Huwe, B., Effect of H and soil structure on transport of sulfonamide antibiotics in agricultural soils. Int. J. Environ. Pollut. 2016, 213, 561-570, DOI: 10.1016/j.envpol.2016.01.089.
• Kang, S. I.; Bae, Y . H., Ph-induced solubility transition of sulfonamide-based polymers. J. CR 2002, 80, 145-155, DOI: 10.1016/S0168-3659(02)00021-4.

In addition, please fix the duplications od the DOI's, i.e.,  “https://doi.org/https://doi.org/...” in the reference section.

Author Response

As I mentioned previously, it is highly desirable practice to cite original pieces of works rather than referring indirectly to them. Therefore, the authors should cite the proper ones when referring to the pKa of SMT as Ref. 25., i.e., 

  • Park, J. Y.; Huwe, B., Effect of H and soil structure on transport of sulfonamide antibiotics in agricultural soils. Int. J. Environ. Pollut. 2016, 213, 561-570, DOI: 10.1016/j.envpol.2016.01.089.
  • Kang, S. I.; Bae, Y . H., Ph-induced solubility transition of sulfonamide-based polymers. J. CR 2002, 80, 145-155, DOI: 10.1016/S0168-3659(02)00021-4.

In addition, please fix the duplications od the DOI's, i.e.,  “https://doi.org/https://doi.org/...” in the reference section.

Response:

Once again, we appreciate the reviewer's valuable comments. We agree that primary works should be cited whenever possible. As we mentioned in the first round of revisions, however, we removed all information related to the pKa of SMT from the manuscript because it could lead to ambiguity in the discussion. Therefore, citing the works suggested by the reviewer is unnecessary. The doi was corrected, and the entire bibliography was revised.